# An Evaluation of Gearbox Condition Monitoring Using Infrared Thermal Images Applied with Convolutional Neural Networks

**DOI:** 10.3390/s19092205

**Published:** 2019-05-13

**Authors:** Yongbo Li, James Xi Gu, Dong Zhen, Minqiang Xu, Andrew Ball

**Affiliations:** 1School of Aeronautics, Northwestern Polytechnical University, Xian 710072, China; 2Centre for Efficiency and Performance Engineering, University of Huddersfield, Queensgate, Huddersfield HD1 3DH, UK; x.gu@hud.ac.uk (J.X.G.); andrew.ball@hud.ac.uk (A.B.); 3School of Mechanical Engineering, Hebei University of Technology, Tianjin 300401, China; d.zhen@hebut.edu.cn; 4Astronautical Science and Mechanics, Harbin Institute of Technology (HIT), No.92 West Dazhi Street, Harbin 150001, China; xumq@hit.edu.cn

**Keywords:** fault diagnosis, infrared thermal imagining, convolutional neural networks, gear faults, vibration method

## Abstract

As an important machine component, the gearbox is widely used in industry for power transmission. Condition monitoring (CM) of a gearbox is critical to provide timely information for undertaking necessary maintenance actions. Massive research efforts have been made in the last two decades to develop vibration-based techniques. However, vibration-based methods usually include several inherent shortages including contact measurement, localized information, noise contamination, and high computation costs, making it difficult to be a cost-effective CM technique. In this paper, infrared thermal (IRT) images, which can contain information covering a large area and acquired remotely, are based on developing a cost-effective CM method. Moreover, a convolutional neural network (CNN) is employed to automatically process the raw IRT images for attaining more comprehensive feature parameters, which avoids the deficiency of incomplete information caused by various feature-extraction methods in vibration analysis. Thus, an IRT–CNN method is developed to achieve online remote monitoring of a gearbox. The performance evaluation based on a bevel gearbox shows that the proposed method can achieve nearly 100% correctness in identifying several common gear faults such as tooth pitting, cracks, and breakages and their compounds. It is also especially robust to ambient temperature changes. In addition, IRT also significantly outperforms its vibration-based counterparts.

## 1. Introduction

The gearbox is widely used for mechanical power transmission in industries such as petroleum equipment, mining machines, chemical industry, and railway applications [1,2]. Because of harsh operating conditions and inherent manufacturing imperfection, gearboxes can be prone to a variety of premature faults and failures. Timely detection and diagnosis of these faults by performing condition monitoring will avoid any catastrophic failures and major economic losses, and ensure their safe operation [3,4].

A variety of signals have been investigated for the condition monitoring (CM) of gearboxes [5,6,7,8,9], including vibration signal [10,11,12], current signal [9], acoustic emission signal [13,14,15], sound signal [16], torque signal [17], and rotating encoder signal [18] and so on. Among these sensor signals, the vibration-based diagnostic method is the most commonly researched because it can directly represent the dynamic behavior of rotating machines [19,20,21].

However, vibration-based methods can have many intrinsic shortages, including requiring contact measurement, providing only localized information, noise contamination, and high computation costs, making it difficult to be a cost-effective CM technique [22]. Moreover, an accelerometer is not allowed in many industrial applications [18]. For these reasons, significant progress has been made in the search for an alternative sensing technique to monitor the health condition of gearboxes.

Recently, temperature signal has gained noticeable attention. The temperature signal is often collected using a thermocouple. Since the fault can cause the temperature of machinery increase, recent research has shown the temperature signal carries rich diagnostic information for CM of rotating machinery [23,24,25]. The thermal camera, as an IRT image-measuring device, can measure the surface temperature of the object without contact. As a result, the infrared thermal (IRT)-based remote CM technique has attracted many researchers’ attention [26]. Lim et al. proposed a fault diagnosis method using IRT images along with support vector machine (SVM) to identify machinery faults [26]. Glowacz et al. introduced a novel method for area selection of IRT image differences as the fault features for fault diagnosis of induction motors [22]. Younus Janssens et al. developed an IRT-based intelligent fault diagnosis method of rotating machinery using discrete wavelet transform, feature selection tool and two classifiers [27]. Janssens et al. combined the three features extracted from IRT images with a SVM to conduct the fault diagnosis of rolling bearings [28]. Meanwhile, the Gini coefficient and machine learning methods are applied for early fault detection of rotating machinery using IRT images by Janssens et al. [29]. From the above literature review, we find that IRT images can provide an alternative and non-invasive way for remote monitoring of rotating machinery [30].

However, the feature-extraction methods mentioned above are handcrafted. The feature designing requires a considerable amount of engineering skills and domain expertise, which significantly influences the description ability and final classification results. In particular, when the IRT images become more challenging, the representational ability of those features may become limited or even powerlessness.

To overcome the shortages of handcrafted features, deep-learning features, which are automatically trained from images using deep-learning methods, are considered to be a more feasible strategy. The key advantage of deep-learning methods is that the deep-learning features are automatically taught from images, instead of relying on manually designed features. Convolutional neural network (CNN) is an effective deep-learning method to extract the features of raw data automatically [31]. CNN can extract the features from the images directly, which avoids the information loss brought by artificial processing. Meanwhile, convolution and pooling are the key operations of CNN, in which convolution is used to extract local image features and pooling is employed to reduce data dimension. Therefore, CNN has far fewer connections and parameters, which results in a system that is easier to train [32,33]. In this paper, CNN is employed for fault-feature extraction.

After the fault-feature extraction using CNN, a classifier is usually required to classify different fault types. In this paper, SoftMax regression (SR) is trained to automatically identify various health conditions. SR is an expansion of logistic regression for multi-classification problems, which has been applied in many fields such as speech emotion recognition [34], facial emotion recognition [35], and text classification [36]. Moreover, SR has a higher calculation efficiency, which is easy to implement [37,38,39].

Based on these distinctive merits of IRT images and CNN, a novel method of using IRT images with CNN and SR, abbreviated to IRT–CNN, is proposed in this study to achieve online remote CM of gearboxes. To demonstrate the superiority of using IRT images, the vibration signal is also implemented using CNN method (denoted as Vib-CNN). The rest of this paper is organized as follows: In Section 2, the theoretical background of CNN and SR is reviewed. In Section 3, the fundamental of SR is briefly introduced. In Section 4, the steps of the proposed IRT–CNN method are given. In Section 5, the proposed method is used to classify types of various common gear faults. Finally, Section 6 draws a conclusion.

## 2. CNN-Based Feature Extraction

CNN is a type of feed-forward neural network, which consists of several filter stages and one classification stage. The filter stages are mainly composed of four different layers: convolutional layer, batch normalization layer, activation layer, and pooling layer. The classification stage consists of several fully connected layers and followed by a classification layer. Figure 1 illustrates the basic architecture of a CNN in processing IRT images.

### 2.1. Convolutional Layer

The convolutional layer convolves the input local regions with a series of filter kernels and generates new features. Each filter uses the same kernel to extract the local feature of the inputs (known as weight-sharing). Here, we use $k_{l}\in R^{J\times D\times H}$ represents the filter kernel in layer *l*, where J,D,H indicate the number, the depth, and the height of the kernels, respectively. The convolutional operation is defined as follows.
(1)$$x_{j}^{l}=k_{j}^{l}*x_{j}^{l-1}+b_{j}^{l}$$
where $b_{j}^{l}$ represents the bias vector and $x_{j}^{l}$ represents the *j*th output feature vector in layer *l*. The notation * represents the dot product operation.

### 2.2. Batch Normalization Layer

Batch normalization layer (BNL) aims to reduce the shift of internal covariance and accelerate the training process of the deep neural network. BNL is often added after the convolutional layer and before the activation layer. The batch normalization process can be expressed as:
(2)$$x^{l-1}=\frac{x^{l-1}-\mu_{B}}{\sqrt{\sigma_{B}^{2}+\varepsilon }}$$
(3)$$x^{l}=\gamma^{l}x^{l-1}+\beta^{l}$$
where $\mu_{B}=E\left[x^{l-1}\right]$, $\sigma_{B}=Var\left[x^{l-1}\right]$. ε denotes a small constant added for numerical stability. $\gamma^{l}$ and $\beta^{l}$ denote the scale and shift parameters to be learned, respectively.

### 2.3. Activation Layer

Activation layer can improve the representation ability of the CNN with a nonlinear mapping function. In this paper, Rectified Linear Unit is employed to accelerate the convergence of the CNN. The output of activation layer can be expressed as:
(4)$$x^{l}=\max\left\{x^{l-1},0\right\}$$


### 2.4. Pooling Layer

Pooling layer can be treated as a down-sampling operation to reduce the size of the features and parameters of the network. Max-pooling layer is commonly used to extract the maximum of the input feature. The max-pooling process can be expressed in Equation (Equation 5).
(5)$$x^{l}=\max_{(j-1)W+1\leq i\leq jW}\left\{x^{l-1}\left(i\right)\right\}$$
where *W* denotes the width of the max-pooling region.

## 3. SoftMax Regression

To identify and diagnose multiple fault types (more than 2 types) which often can occur in gearboxes, SoftMax regression (SR) is adopted as the classifier after the final layer in CNN method. SR is an expansion of logistic regression for multi-classification problems. By estimating the probability of one sample belonging to each category label, SR guarantees better classification performance [38]. Meanwhile, SR is easy to implement with high computation efficiency [38]. Figure 2 illustrates a SR classification model.

Let $D=\{\left(x^{\left(1\right)},y^{\left(1\right)}\right),\cdots ,\left(x^{\left(n\right)},y^{\left(n\right)}\right)\}$ represent the training set, $x^{\left(i\right)}\left(i\in \left\{1,2,\cdots ,n\right\}\right)$ represent the training data, and $y^{\left(j\right)}\left(j\in \left\{1,2,\cdots ,n\right\}\right)$ represent the corresponding health condition label. SR aims to estimate the probability p(y=k|x;θ) for the training data $x^{\left(i\right)}\left(i\in \left\{1,2,\cdots ,n\right\}\right)$ belongs to each health condition label $y^{\left(j\right)}\left(j\in \left\{1,2,\cdots ,n\right\}\right)$ The probability p(y=k|x;θ) can be obtained from the hypothesis function $h_{\theta }\left(x\right)$ as follows:
(6)$$h_{\theta }\left(x^{\left(i\right)}\right)=\left[\begin{array}{c} p(y^{\left(i\right)}=1|x^{\left(i\right)};\theta )\\ p(y^{\left(i\right)}=2|x^{\left(i\right)};\theta )\\ \vdots \\ p(y^{\left(i\right)}=k|x^{\left(i\right)};\theta ) \end{array}\right]=\frac{1}{\sum_{j=1}^{k}e^{\theta_{j}^{T}x^{\left(i\right)}}}\left[\begin{array}{c} e^{\theta_{1}^{T}x^{\left(i\right)}}\\ e^{\theta_{2}^{T}x^{\left(i\right)}}\\ \vdots \\ e^{\theta_{k}^{T}x^{\left(i\right)}} \end{array}\right]$$
where the output value of the hypothesis function $h_{\theta }\left(x\right)$ is a *k* dimensional vector $\theta_{1},\theta_{2},\cdots ,\theta_{k}$. It should be noted that the $\sum_{j=1}^{k}e^{\theta_{j}^{T}x^{\left(i\right)}}$ normalizes the $e^{\theta_{1}^{T}x^{\left(i\right)}}$ thereby, the summation of the elements equals 1.

Based on the hypothesis function $h_{\theta }\left(x\right)$, SR model is trained through successive adjustments to minimize the loss function of Equation (Equation 7).
(7)$$J\left(\theta \right)=-\frac{1}{m}\left[\sum_{i=1}^{m}\sum_{j=1}^{k}1\left\{y^{\left(i\right)}=j\right\}\log\frac{e^{\theta_{j}^{T}x^{\left(i\right)}}}{\sum_{l=1}^{k}e^{\theta_{l}^{T}x^{\left(i\right)}}}\right]$$
where 1{.} represents the indicator function. If the condition is true, returning 1, otherwise, returning 0.

## 4. IRT–CNN Method for Gearbox Diagnosis

By combining the capabilities of CNN and SR, it is relatively straightforward to construct a procedure to automatically identify IRT images and differentiate various gear fault cases. It consists mainly of two learning phases as shown in Figure 3. The fittings phases are for applying CNN to raw IRT images to extract the discriminative features, which are followed by a SR with the features as input to classify the types of gear faults using the features. It is noteworthy that no expert knowledge of failure mechanisms and parameter setting are required beforehand for applying the proposed method, which makes it easy to implement in real applications.

To demonstrate the implementation of IRT–CNN method, this study follows a procedure as shown in Figure 3, which may be described with five sequential steps starting with the bottom block of Figure 3:

Step 1: Collect the IRT video using the thermal camera with sufficient accuracy, for example, as specified in Table 1;

Step 2: Select IRT images under each health condition of gearboxes;

Step 3: Divide the images into training samples and testing samples;

Step 4: Apply CNN to training samples to obtain a set of deep-learning features, which is then fed to SR to classify fault types;

Step 5: Validate the classification performance of proposed IRT–CNN-based fault diagnosis method using the thermal images collected from a bevel gearbox.

## 5. Experimental IRT Images Acquisition

### 5.1. Experimental Setup

To verify the method, a series of IRT images were collected from a gearbox on SpectraQuest Machinery Fault Simulator (MFS) illustrated in Figure 4. The schematic diagram of the test rig and thermal camera are shown in Figure 4b. The test simulator consists of a three-way gearbox with straight-cut bevel gears, which is driven by a reliance electric motor and loaded under 0.56 Nm (or 5 in-lbs) by a magnetic clutch on the rear of the gearbox. The pinion gear has 18 teeth and the transmission ratio is 1.5. For a comparative study, an accelerometer was also installed on the top of the gearbox to collect the vibration signals with a sampling frequency of 12,800 Hz. During testing, the motor speed was set to a constant speed of 3000 rpm. These steady operating conditions are common in many industrial applications such as pumps and compressors in a petrochemical process.

In this experiment, a Hawk-1384 thermal camera is employed, and its corresponding parameters are shown in Table 1. To eliminate the influence of heat transfer path, a germanium glass (diameter: 3 cm) is installed on the top of the gearbox to directly acquire the IRT images of gears and shaft because the infrared ray can pass through the germanium glass.

To show the performance of proposed method, eight gear-health conditions are designed to simulate various fault conditions occurring in a gear transmission, which include pitting in the driving tooth (PT), a broken tooth in the driving gear (BT), a missing tooth in the driving gear (MT), crack in the follower tooth (CT), pitting in the driving gear with crack in the follower tooth (PC), broken tooth in the driving gear with crack in the follower tooth (BC), missing tooth in the driving gear with crack in the follower tooth (MC). The details of gear faults are illustrated in Figure 5, which are typical scenarios for early stages of faulty gearbox.

For each health condition of gearbox, the temperature influence was considered, thereby, the IRT videos were recorded throughout the entire temperature-rising process using the thermal camera. The experiment is terminated when the gearbox comes into a steady stage [29,40]. To determine whether the temperature of gearbox reaches a stable condition, a circle around the gear meshing area is added as a reference, in which the high-, average-, and low-temperature values can be observed, as shown in Figure 6a. Please note that there is no marked area using the circle and corresponding color bar in the pseudo-colored frame extracted from IRT video, as shown in Figure 6b. It is worth noting that we conduct only one run of the machine for each fault type. Then, the IRT images are extracted from the obtained IRT videos. The detailed steps for the thermal image acquisition are illustrated in Algorithm 1.

**Algorithm 1** Steps for the thermal image acquisition.
Step 1: Select one fault type from the ten health conditions.Step 2: Initialize the temperature of MFS system to the environment temperature (18.9 °C in this paper).Step 3: Conduct the thermal video acquisition under a constant speed of 3000 rpm.Step 4: Observe the highest temperature through the IRT camera (around the testing bearing). When the highest temperature reaches a relatively stable temperature (60 °C in this test), stop the thermal video acquisition.Step 5: Cool down the MFS system to the environment temperature.Step 6: Repeat the above steps for another fault type until the end of experiment.Step 7: Extract the thermal images from the thermal videos under each health condition.


### 5.2. Condition Monitoring Performance Evaluation

To evaluate the performance of IRT–CNN, the image data is organized into two typical scenarios, which are the case of a specified temperature value and a specified temperature range based on the high-temperature values acquired in the highlighted area of Figure 6. These scenarios not only allow the evaluation to be archived under different difficulty levels of data diversity but also exemplify the details of implementing the method.

The specified temperature case, denoted as Scenario 1, will use the temperature distribution differences caused by the various faults to differentiate the cases. The specified temperature ranges such as 39 °C–42 °C, denoted as Scenario 2, will use differences in both distribution and absolute values to differentiate the faults, taking into account the influences of inevitable fluctuations of temperatures caused by ambient temperatures.

#### 5.2.1. Evaluation with IRT Image at a Specified Temperature

Scenario 1 tests the performance of the proposed method using the IRT images when the gearbox reaches its steady-state operation, i.e., the casing temperature rises to and stabilizes at 54 °C in this paper. The IRT system will collect data from 8 gears with different health conditions. Each condition has 100 image samples. In total, there are 800 samples (100 samples × 8 health conditions). 35% of samples of these samples are used to train the proposed IRT–CNN, and the rest samples are used to test the performance of the proposed method. The detailed descriptions of sample numbers for Scenario 1 are given in Table 2.

The raw IRT images of gearbox under eight gears with different health conditions are illustrated in Figure 7. It is hard to distinguish the fault types visually though observing the IRT images as the difference of IRT image between each of the health conditions are very small. Thus, CNN is employed to extract features from raw IRT images. The parameter settings of CNN are listed in Table 3. It contains four convolutional layers (CL1, CL2, CL3 and CL4) and two fully connected layers (F1 and F2). The filter size is 5×5 for the first two convolutional layers and 3×3 for the remaining two layers. Following each convolutional layer, a 2×2 max-pooling layer is carried out. The F1 and F2 layers have 1024 and 512 units, respectively. At the end, an eight-way SR classifier is used to identify the health conditions. The classification results are illustrated in Figure 8. It shows that the classification rate is 100% right i.e., no testing samples is misclassified. This validates that the proposed IRT–CNN method is effective in recognizing the types of various common gear faults based on the IRT images at a given temperature.

In addition, the classification performance of the proposed IRT–CNN method is also examined by using various percentages of training samples. To reduce the effect of randomness, 20 trails are conducted for each experiment. The average training and testing accuracies are calculated, and their corresponding standard deviations are represented using the positive error bars, as shown in Figure 9. It can be observed that the testing accuracy increased, and its standard deviation decreased as more training samples are used. In addition, the average of classification accuracies of the proposed method is 100% with no standard deviation using 35% of samples for training, showing that 35% training, in particular, using 35 training images, will be sufficient to train the CNN for achieving a highly satisfactory accuracy rate.

#### 5.2.2. Comparison with Vibration Signals

To show the superiority of IRT-based remote monitoring technique, a comparison is conducted between IRT–CNN and Vib-CNN. Figure 10 and Figure 11 display the time domain waveforms of vibration signals and their corresponding envelope spectra of gearbox, respectively. For fair comparison, the Vib-CNN method also adopts 35% samples for training and 65% samples for testing. The classification results of Vib-CNN method are displayed in Figure 12 and Table 4, respectively. It can be observed from Table 4 that the classification accuracy of Vib-CNN is at 71.53% (372/520), which is much lower than that of the proposed IRT–CNN method. The comparison results demonstrate that the proposed IRT–CNN method outperforms Vib-CNN method significantly. Meanwhile, the CPU time is recorded to measure the calculation complexity of the two methods with the results given in Table 4. A desktop with 3.4 GHz i7-Core CPU, 16.0 GB RAM and Python 3.6 is used to run the code. It can be found from Table 4 that IRT–CNN method takes slightly longer than Vib-CNN method.

Furthermore, to show the features extracted from IRT images and vibrations, three-dimensional projections are used for visualizing using PCA, as shown in Figure 13. It can be observed that the samples are all distributed around the class center and the samples are discriminated clearly using proposed IRT–CNN method, as shown in Figure 13a. This means that the extracted features from IRT–CNN has the distinguishable and robust information. However, as shown as shown in Figure 13b. The vibration features extracted by CNN are less clustered along with a much wider spread, showing that it is hard to achieve satisfactory result classification. This further demonstrates the IRT images provide wealthy and consistent information regarding the gear’s health conditions.

Figure 14 shows the overall accuracy curves of the training and testing for proposed IRT–CNN and Vib-CNN methods over 100 epochs. As seen in Figure 14, it can be clearly observed that both the training accuracy and testing accuracy of proposed IRT–CNN method converge to a stable value after 30 epochs, which is faster than that of Vib-CNN method after 50 epochs. This phenomenon demonstrates that the proposed IRT–CNN method has a better convergence performance compared with Vib-CNN method.

#### 5.2.3. Evaluation with IRT Images from Specified Temperature Ranges

In Scenario 2, the performance of the proposed IRT–CNN method is first examined under six small temperature ranges, each of them covering an interval of 3 °C, to investigate the influences of temperature fluctuations. These six temperature ranges are Range 1 (36 °C–39 °C), Range 2 (39 °C–42 °C), Range 3 (42 °C–45 °C), Range 4 (45 °C–48 °C), Range 5 (48 °C–51 °C), and Range 6 (51 °C–54 °C), which are sorted by the temperatures in the gear meshing area. As an example, Figure 15 shows the IRT images of the healthy gear in the six ranges. A careful observation can note that there is a gradual temperature change in the zone highlighted by dashed line zone across all six ranges, even though each image was automatically scaled when the imagery was captured. Evidently, the diversity of temperature fluctuations can induce more challenges to the deep-learning process for accurate classification.

Each temperature range consists of 800 samples (100 samples × 8 gear-health conditions). 35% samples (280 data samples) are taken as the training samples and the remaining 65% data as the testing samples. Also, 20 trails were conducted to reduce the effect of randomness. Figure 16 and Table 5 show the obtained final classification results for Scenario 2. It can be seen that the testing accuracy of proposed IRT–CNN decreases by only 4.03% changing from 100% to 95.97% using this data set. The comparison results indicate the proposed IRT–CNN method also yields satisfactory identification results under small temperature fluctuations in the environment.

To better understand the classification results under small temperature fluctuations, a temperature range of 39 °C–42 °C is selected as an example due to its lowest training and testing accuracies. Figure 17 illustrates the confusion matrix results of proposed IRT–CNN method in the temperature range of 39 °C–42 °C. As seen in Figure 17, it can be observed that there are 8 testing samples with pitting tooth fault are misclassified into missing tooth fault with the final testing accuracy of 87.69%. This phenomenon can be explained in the following way. Since the temperature distributions of eight health conditions are not stable as the temperature goes up, this will cause the measured IRT images of eight health condition are difficult to be distinguished.

Furthermore, images with a wider temperature range of 36 °C–54 °C are also investigated to evaluate the performance of proposed IRT–CNN. In total there are 4800 samples (800 samples × 6 different temperature ranges) aggregated, 35% samples (1680 samples) of which are taken for training and the rest 65% samples (3120 samples) for testing. The classification results are listed in Table 5. As seen from Table 5, the identification of proposed IRT–CNN method reduces to 95.53% with a standard deviation of 0.912. The classification results show that proposed IRT–CNN method still produces a satisfactory output even when operating in a much wider range of temperature fluctuations. This means that the proposed method is robust to ambient changes and verified to be accurate and reliable for different applications.

## 6. Conclusions

This paper develops a novel method for the CM of gearboxes using IRT images with CNN deep-leaning method (IRT–CNN). It combines the remote measurement merits of IRT images and automatic processing of CNN so that it can be more cost-effective and easily implemented. The effectiveness and reliability of the proposed IRT–CNN method is demonstrated using the IRT images collected from a bevel gearbox test rig. Results demonstrate that the proposed IRT–CNN is sensitive and reliable in recognizing several common gear faults, achieving a correct classification rate of 100% for IRT images acquired at specified temperatures of 95.97%, and 95.53% for the IRT with temperature ranges of 3 °C and 19 °C, replicating potential ambient temperature changes in real operational environments. Moreover, it also found that vibration signals with CNN can produce an identification rate of 71.53% for the constant temperature scenario, which is much lower that of IRT–CNN.

In this preliminary study, the proposed method has been demonstrated to be a promising tool for fault classification of a bevel gearbox using temperature signals. However, for other type machinery, such as motor faults, the effectiveness of the proposed method is unknown. Future work includes more validations on other machinery in a more realistic environment and deep analysis of fault mechanism using temperature signal.

## Figures and Tables

**Figure 1 sensors-19-02205-f001:**
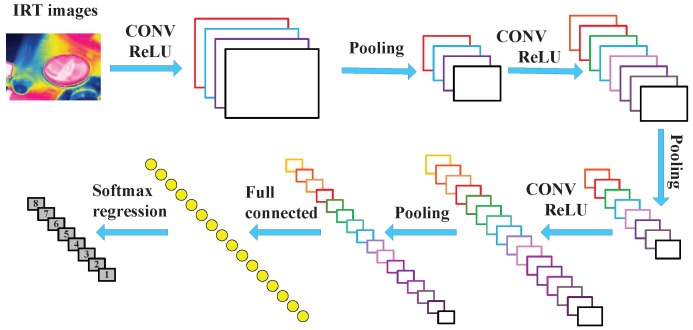
Architecture of the convolutional neural networks.

**Figure 2 sensors-19-02205-f002:**
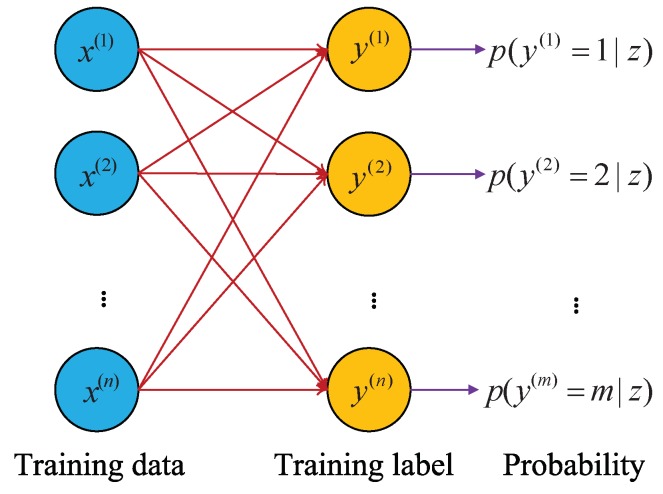
SoftMax regression model.

**Figure 3 sensors-19-02205-f003:**
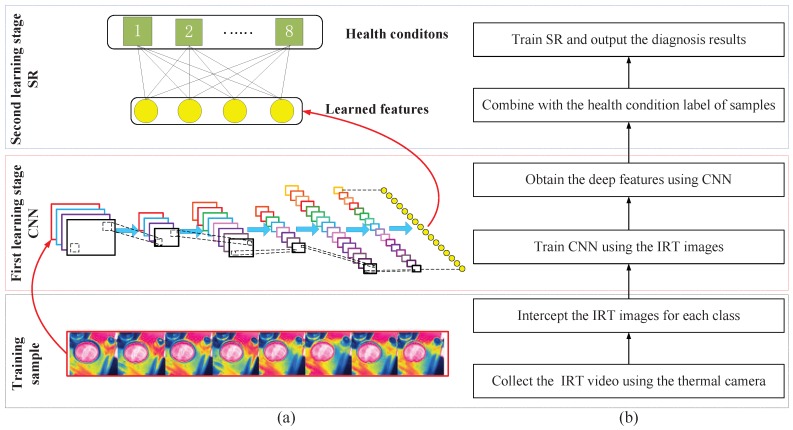
Flowchart of the proposed IRT–CNN method for fault diagnosis of gearbox: (**a**) illustrative diagrams, and (**b**) flowchart of training process.

**Figure 4 sensors-19-02205-f004:**
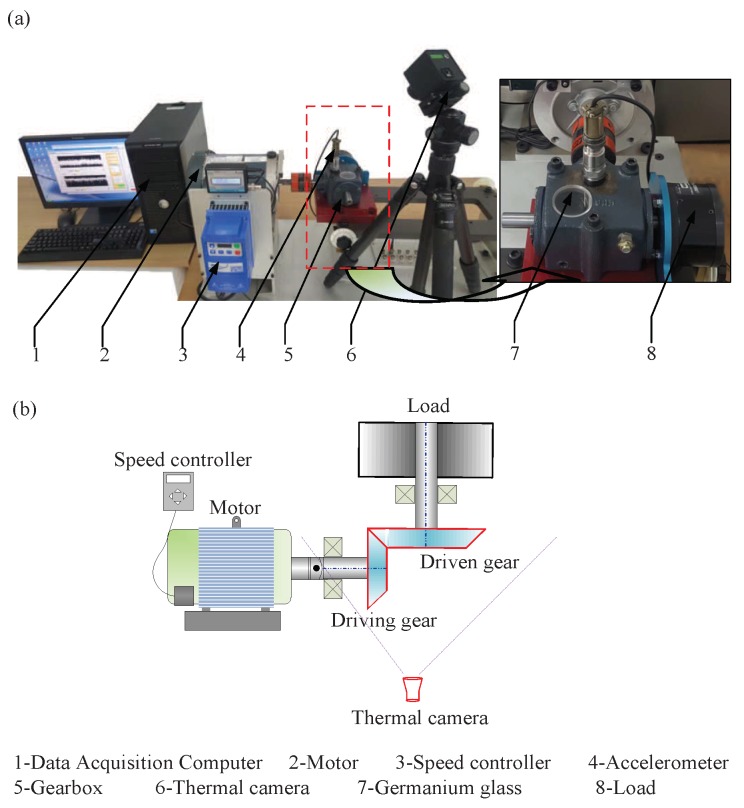
(**a**) The machinery fault simulator system, (**b**) the schematic diagram of the test rig and thermal camera setup.

**Figure 5 sensors-19-02205-f005:**
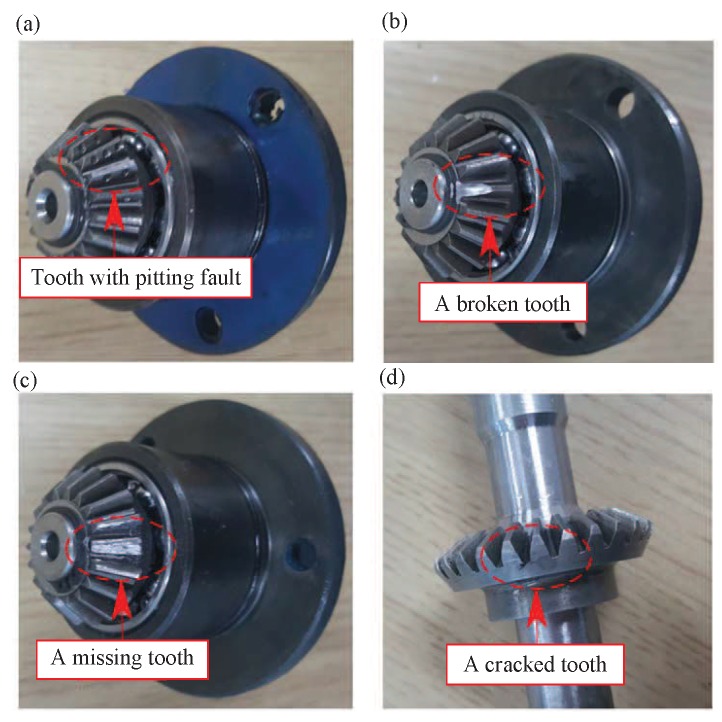
The designed gear faults: (**a**) Pitting in the driving tooth; (**b**) A broken tooth of driving gear; (**c**) A missing tooth of driving gear; (**d**) A cracked tooth of follower gear.

**Figure 6 sensors-19-02205-f006:**
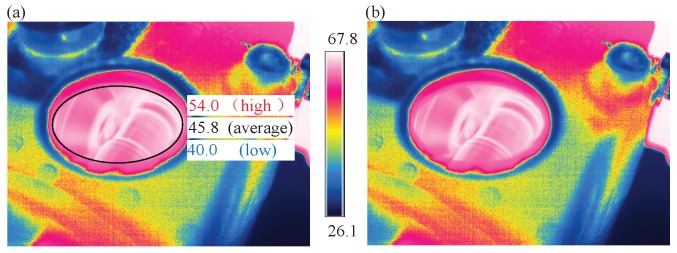
IRT images extracted from an infrared thermal video: (**a**) marked meshing area for reference (**b**) the raw IRT images.

**Figure 7 sensors-19-02205-f007:**
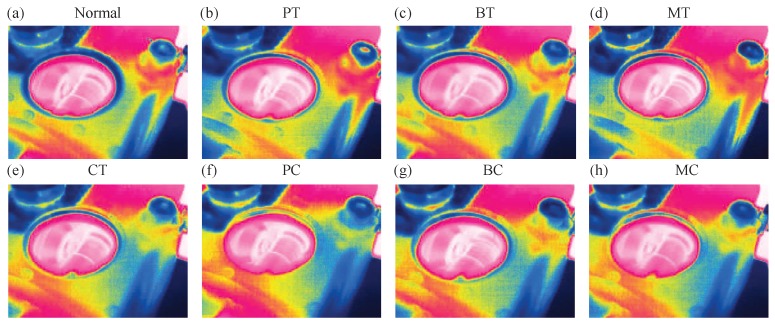
Thermal images under eight gears with different health conditions: (**a**) healthy condition (**b**) pitting in the driving tooth (**c**) a broken tooth in the driving gear (**d**) a missing tooth in the driving gear (**e**) crack in the follower tooth (**f**) pitting in the driving gear with crack in the follower tooth pitting teeth (**g**) broken tooth in the driving gear with crack in the follower tooth (**h**) missing tooth in the driving tooth with crack in the follower tooth.

**Figure 8 sensors-19-02205-f008:**
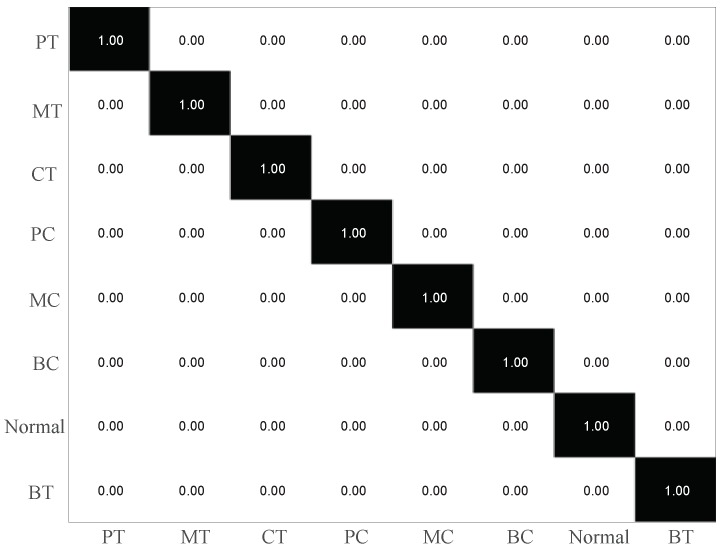
The classification results of the IRT–CNN method.

**Figure 9 sensors-19-02205-f009:**
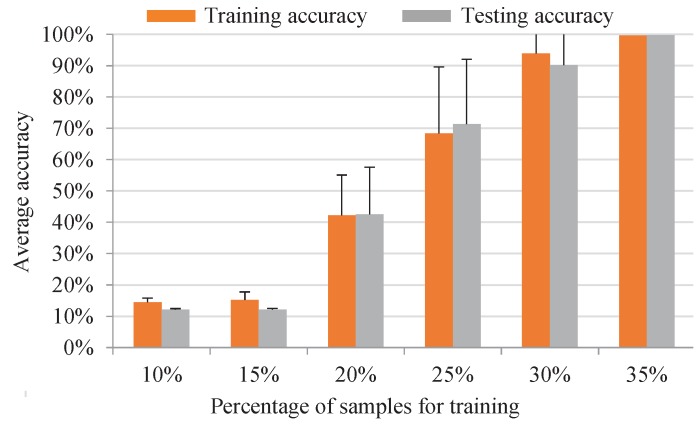
The classification results of the IRT–CNN trained by different percentages of data samples.

**Figure 10 sensors-19-02205-f010:**
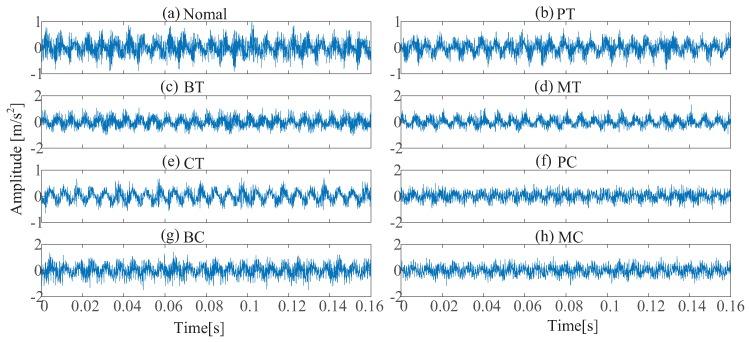
The vibration signals for eight gear-health conditions: (**a**) healthy condition (**b**) pitting in the driving tooth (**c**) a broken tooth in the driving gear (**d**) a missing tooth in the driving gear (**e**) crack in the follower tooth (**f**) pitting in the driving gear with crack in the follower tooth (**g**) broken tooth in the driving gear with crack in the follower tooth (**h**) missing tooth in the driving tooth with crack in the follower tooth.

**Figure 11 sensors-19-02205-f011:**
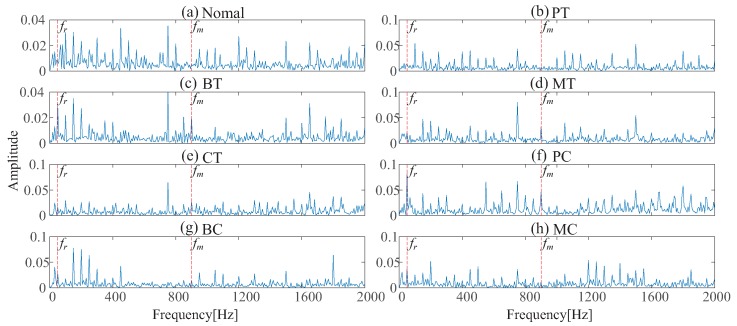
Envelope spectra for eight gear-health conditions: (**a**) healthy condition (**b**) pitting in the driving tooth (**c**) a broken tooth in the driving gear (**d**) a missing tooth in the driving gear (**e**) crack in the follower tooth (**f**) pitting in the driving gear with crack in the follower tooth (**g**) broken tooth in the driving gear with crack in the follower tooth (**h**) missing tooth in the driving tooth with crack in the follower tooth.

**Figure 12 sensors-19-02205-f012:**
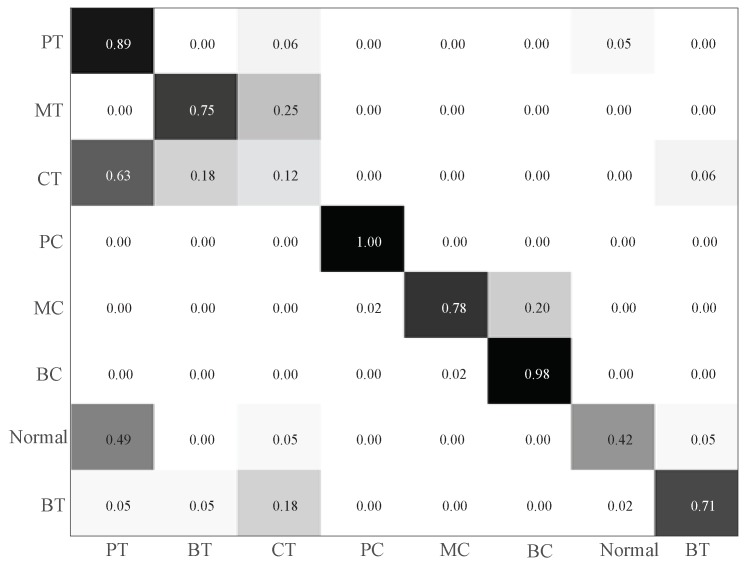
The classification results using Vib-CNN method.

**Figure 13 sensors-19-02205-f013:**
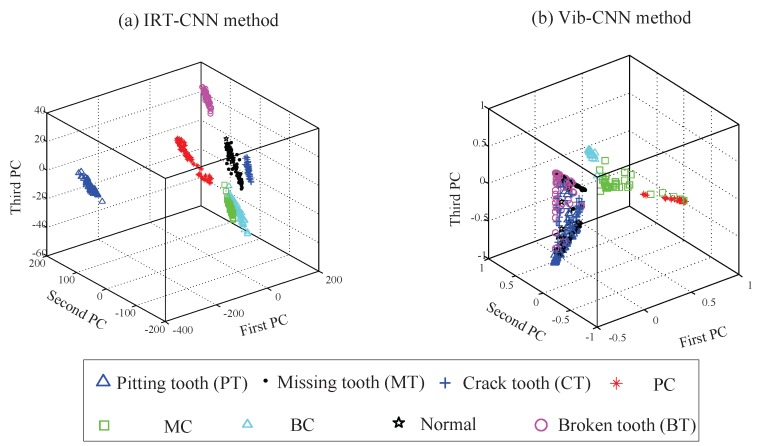
Projections of the features using PCA with different source signals: (**a**) IRT –CNN method (**b**) Vib-CNN method.

**Figure 14 sensors-19-02205-f014:**
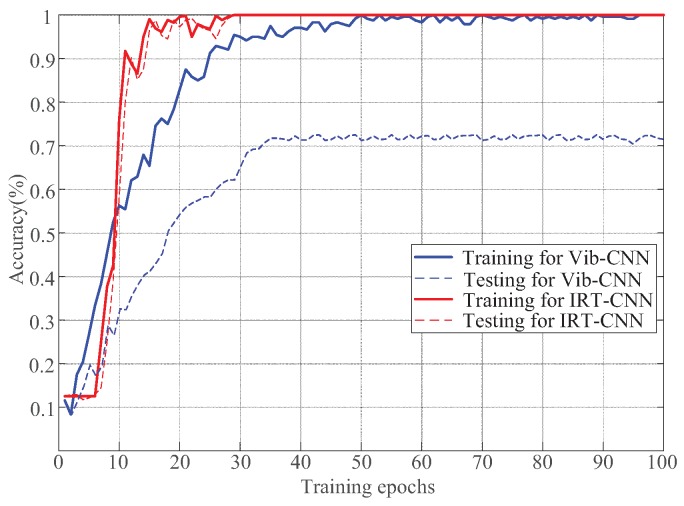
Training and testing performance curves in terms of overall accuracy.

**Figure 15 sensors-19-02205-f015:**
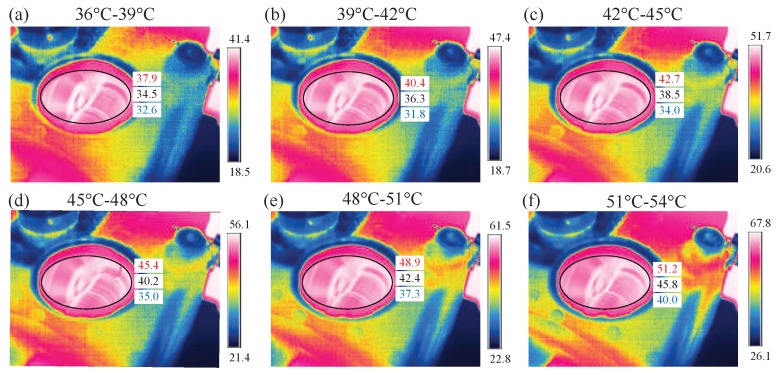
IRT images under normal condition of gearbox in six temperature ranges: (**a**) 36 °C–39 °C (**b**) 39 °C–42 °C (**c**) 42 °C–45 °C (**d**) 45 °C–48 °C (**e**) 48 °C–51 °C (**f**) 51 °C–54 °C.

**Figure 16 sensors-19-02205-f016:**
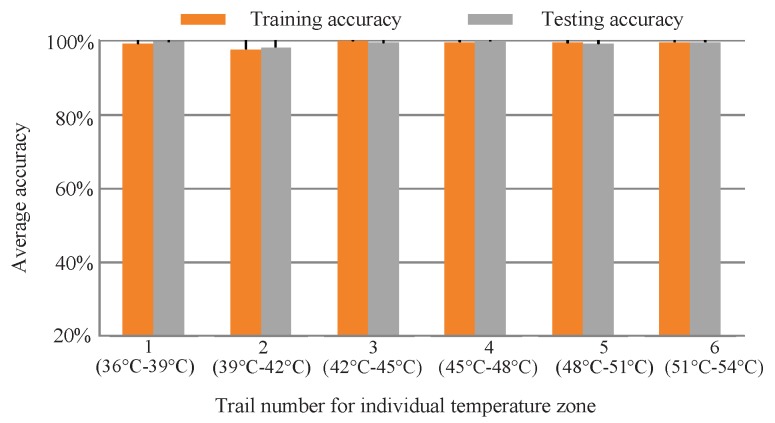
Diagnosis results for each temperature range using proposed IRT–CNN method.

**Figure 17 sensors-19-02205-f017:**
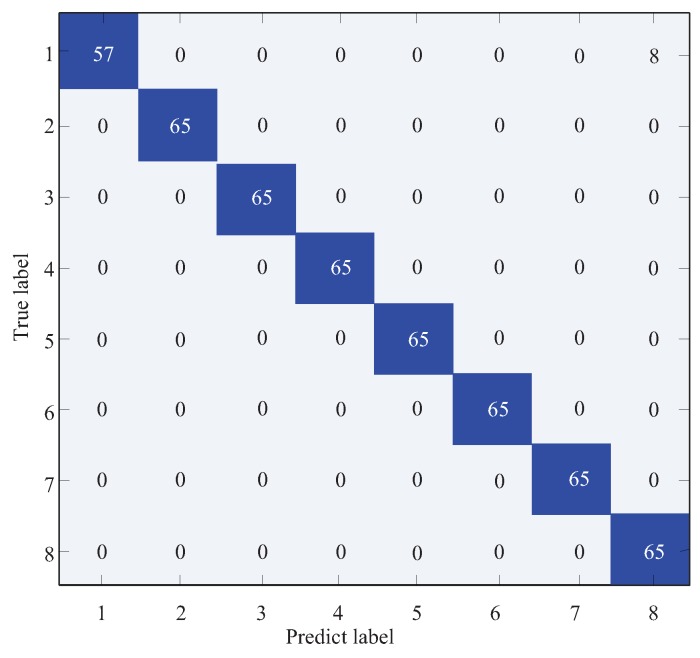
Confusion matrix results with a temperature range of 39 °C–42 °C.

**Table 1 sensors-19-02205-t001:** Configuration parameters setting of the thermal camera.

Configuration Parameters	Values
Producer of thermal camera	Hawk, China
Image resolution	384 × 288
Frame rate	25 fps
Temperature measurement range	−25 °C 260 °C
Environmental temperature	18.9 °C
Thermal sensitivity	0.05 °C
Palette	rainbow
Contrast	50
Brightness	50
Gain	2

**Table 2 sensors-19-02205-t002:** The detailed description of sample numbers for Scenario 1.

Fault Types	Class Label	Number of Training Data	Number of Resting Data
PT	1	35	65
BT	2	35	65
CT	3	35	65
PC	4	35	65
MC	5	35	65
BC	6	35	65
Normal	7	35	65
MT	8	35	65

**Table 3 sensors-19-02205-t003:** The detailed structure of CNN.

Layer Type	Number of Filter	Size of Feature Map	Size of Kernel	Number of Stride	Number of Padding
Image input layer	-	100 × 100 × 3	-	-	-
CL1 (Convolution layer-1)	32	100 × 100 × 32	5 × 5	1 × 1	2 × 2
M1 (Max-Pooling layer-1)	1	50 × 50 × 32	2 × 2	2 × 2	0 × 0
CL2 (Convolution layer-2)	64	50 × 50 × 64	5 × 5	1 × 1	2 × 2
M2 (Max-Pooling layer-2)	1	25 × 25 × 64	2 × 2	2 × 2	0 × 0
CL3 (Convolution layer-3)	128	25 × 25 × 128	3 × 3	1 × 1	1 × 1
M3 (Max-Pooling layer-3)	1	12 × 12 × 128	2 × 2	2 × 2	0 × 0
CL4 (Convolution layer-4)	128	12 × 12 × 128	3 × 3	1 × 1	1 × 1
M4 (Max-Pooling layer-4)	1	6 × 6 × 128	2 × 2	2 × 2	0 × 0
F1 (Full connection layer-1)	-	1024 × 1	-	-	-
F2 (Full connection layer-2)	-	512 × 1	-	-	-
Output layer	-	8 × 1	-	-	-

**Table 4 sensors-19-02205-t004:** Diagnosis results for Scenario 1 using IRT–CNN and Vib-CNN methods.

Data Source	Average Testing Accuracy	Standard Deviation	CPU Time (s)
Vibration signals	71.53% (372/520)	0.5591	470
IRT images	100% (520/520)	0.00	542

**Table 5 sensors-19-02205-t005:** Diagnosis results for different temperature ranges.

Temperature Ranges	Average Testing Accuracy	Standard Deviation
3 °C range	95.97% (2994/3120)	1.0105
19 °C range	95.53% (2980/3120)	0.912

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
