# Peer review of "An Evaluation of Gearbox Condition Monitoring Using Infrared Thermal Images Applied with Convolutional Neural Networks"

_sensors, 2019, doi:10.3390/s19092205_

Round 1

Reviewer 1 Report

The aim of this paper is to demonstrate how deep learning using CNNs can be applied to predicting failures in gearboxes, using temperature as an input (and also, as a comparison, vibration).

The paper begins with a clear motivation for both why the problem is important, and gives a good prima facie motivation for the approach taken in the paper (in particular, emphasising that because the new method doesn't need features to be hand-crafted, it can be adapted much more readily to a new situation. 

The paper gives a competent review of the CNN methods and how they work, and then explains how they have been applied to this specific problem. This is explained in a level of detail that would enable a knowledgable reader of the paper to implement the algorithm.

The experimental setup, and how the different failures were made, is clearly explained; again, someone who wanted to replicate or extend these experiments would be readily able to do so. One thing I would like to see clarified is how the different failures were conducted; were the 100 repeats all different runs of the machine with the same physical component (e.g. for the "missing tooth" failure, was the same gear with the same tooth missing used each time). this would be useful to know.

The analysis of the results is well-presented. I wondered at first whether it is "fair" to choose the 35% proportion for training data based on using the testing data, but in the end this is presented at the end of the experiment and so this seems fine to me. The comparison with the vibration signals is a good experiment to do, also using the CNN method. It would be have been good to see a comparison against a non-CNN method to try and quantify how much of an improvement CNN makes , but I would not insist on it. It is good to see an analysis of why the new method works, and also some experiments with different experimental setups (temperature). The new method works very well and is likely to be of good practical significance.

One question for the longer term - which clearly I don't think needs investigating for this paper, but which might be good to mention in future work, is whether a more general system could learn to diagnose failures from a number of different machines, and then be applied to a previously unseen machine.

The paper is well written in decent scientific English, and is very clearly structured. The paper makes good use of figures, charts and tables in presenting the work.

Overall, this is a good paper; there are a few minor issues to be addressed, but it offers a well-explained, practical solution to an important problem.

Author Response

Response to Reviewers’ Comments

Sensors-498570: “An evaluation of gearbox condition monitoring using infrared thermal images applied with convolutional neural networks” by Li, Gu, Zhen, Xu and Andrew.

We would like to thank the reviewers for their constructive and helpful comments and suggestions to improve the quality of this paper. We have addressed the reviewers’ comments and queries thoroughly. A summary of our response to the suggested changes from reviewers is given below.

Reviewer #1:

The aim of this paper is to demonstrate how deep learning using CNNs can be applied to predicting failures in gearboxes, using temperature as an input (and also, as a comparison, vibration).

The paper begins with a clear motivation for both why the problem is important, and gives a good prima facie motivation for the approach taken in the paper (in particular, emphasising that because the new method doesn't need features to be hand-crafted, it can be adapted much more readily to a new situation.

The paper gives a competent review of the CNN methods and how they work, and then explains how they have been applied to this specific problem. This is explained in a level of detail that would enable a knowledgable reader of the paper to implement the algorithm.

1. The experimental setup, and how the different failures were made, is clearly explained; again, someone who wanted to replicate or extend these experiments would be readily able to do so. One thing I would like to see clarified is how the different failures were conducted; were the 100 repeats all different runs of the machine with the same physical component (e.g. for the "missing tooth" failure, was the same gear with the same tooth missing used each time). this would be useful to know.

Response: Thanks for your comments. There are seven different fault types are considered in this experiment. For each fault type, we conduct only one time of the machine. We record the infrared thermal video using the thermal camera. Then, we extract the IRT images from the IRT videos. As you suggested, we have added more explanations as follows:

To show the performance of proposed method, seven gear faults are designed to simulate various fault conditions occurring in a gear transmission, which include pitting in the driving tooth (PT), a broken tooth in the driving gear (BT), a missing tooth in the driving gear (MT), crack in the follower tooth (CT), pitting in the driving gear with crack in the follower tooth (PC), broken tooth in the driving gear with crack in the follower tooth (BC), missing tooth in the driving gear with crack in the follower tooth (MC). The details of gear faults are illustrated in Fig.5, which are typical scenarios for early stages of faulty gearbox.

It is worth noting that we conduct only one run of the machine for each fault type. Then, the IRT images are extracted from the obtained IRT videos. The detailed steps for the thermal image acquisition are illustrated in Algorithm 1.

Algorithm 1: Steps for the thermal image acquisition

Step 1: Select one fault type from the eight health conditions;

Step 2: Initialize the temperature of MFS system to the environment temperature, which was 18.9°C in this study;

Step 3: Operate the MFS system at the constant speed of 3000rpm for one hour and observe the high temperature through the IRT camera around the gear mesh area highlighted in Fig 6;

Step 4: Conduct the thermal image acquisition through the whole temperature increasing process;

Step 5: Cool down the MFS system to the environment temperature;

Step 6: Repeat the above steps for another fault type until the end of experiment;

Step 7: Extract the IRT images from the IRT videos under each health condition.

The related explanations (highlighted in red) have been added in Section 5.1 of pages 7-8 in the revised paper.

2. The analysis of the results is well-presented. I wondered at first whether it is "fair" to choose the 35% proportion for training data based on using the testing data, but in the end this is presented at the end of the experiment and so this seems fine to me. The comparison with the vibration signals is a good experiment to do, also using the CNN method. It would be have been good to see a comparison against a non-CNN method to try and quantify how much of an improvement CNN makes, but I would not insist on it. It is good to see an analysis of why the new method works, and also some experiments with different experimental setups (temperature). The new method works very well and is likely to be of good practical significance.

Response: Thanks for your comments. In this preliminary study, the proposed method has been demonstrated to be a promising tool for fault classification of bevel gearbox using temperature signal. However, for other type machinery, such as motor faults, the effectiveness of the proposed method is unknown. Future work includes more validations on other machinery in a more realistic environment and deep analysis of fault mechanism using temperature signal.

Detailed explanations are added in Section Conclusion of page 14. See the sentences highlighted in red. For your convenience, we recall them as follows:

In this preliminary study, the proposed method has been demonstrated to be a promising tool for fault classification of bevel gearbox using temperature signal. However, for other type machinery, such as motor faults, the effectiveness of the proposed method is unknown. Future work includes more validations on other machinery in a more realistic environment and deep analysis of fault mechanism using temperature signal.

Reviewer 2 Report

The paper presents a method and demonstration of applying CNN-based algorithm to gearbox fault detection in thermal images. The results showed pretty fine accuracies, such as 95%-100%. The paper is well organized for publication. Therefore, I would recommend this for publication in the Sensors. However, as the results are almost perfect, the approach seems attractive though it induces suspicions to unexpected problems in practices. I would like authors to revise the points below.

1. Please add learning curves (accuracy and loss) of the CNN and mention computing time and number of epoch for the convergence.

2. Figure 15 - Please show some examples of the misclassifications and extend discussions on possible noises and errors in the conditions.

3. Some English corrections in my reviews. Please consider extensive proof-reading.

3-a. L26 - Spell out "CM". The abbreviations should be spelled out at the first appearance in body text, not only abstarct.

3-b. L70 - Spell out "SR", which in not defined before this.

3-c. L124 - "..., which is the followed by a SR ..." - remove "the"

Author Response

Response to Reviewers’ Comments

Sensors-498570: “An evaluation of gearbox condition monitoring using infrared thermal images applied with convolutional neural networks” by Li, Gu, Zhen, Xu and Andrew.

We would like to thank the reviewers for their constructive and helpful comments and suggestions to improve the quality of this paper. We have addressed the reviewers’ comments and queries thoroughly. A summary of our response to the suggested changes from reviewers is given below.

Reviewer #2:

The paper presents a method and demonstration of applying CNN-based algorithm to gearbox fault detection in thermal images. The results showed pretty fine accuracies, such as 95%-100%. The paper is well organized for publication. Therefore, I would recommend this for publication in the Sensors. However, as the results are almost perfect, the approach seems attractive though it induces suspicions to unexpected problems in practices. I would like authors to revise the points below.

1. Please add learning curves (accuracy and loss) of the CNN and mention computing time and number of epoch for the convergence.

Response: Thanks for your comments. As you suggest, we have added the overall accuracy curves of the training and testing for proposed IRT-CNN and Vib-CNN methods over 100 epochs, as shown in Fig. 14.  It is worth noticing that the overall loss curves can be deduced through the accuracy curves, which have similar characteristics with accuracy curves. Therefore, for saving space, only the accuracy curves are plotted in this paper.

As seen in Fig.14, it can be clearly observed that both the training accuracy and testing accuracy of proposed IRT-CNN method converge to a stable value after 30 epochs, which is faster than that of Vib-CNN method after 50 epochs. This phenomenon demonstrates that the proposed IRT-CNN method has a better convergence performance compared with Vib-CNN method.

In addition, the CPU time is also recorded to measure the calculation complexity of the two methods with the results given in Table. 4. A desktop with 3.4 GHz i7-Core CPU, 16.0 GB RAM and Python 3.6 is used to run the code. It can be found from Table. 4 that IRT-CNN method takes slightly longer than Vib-CNN method.

Fig.14 Training and testing performance curves in terms of overall accuracy. 

Table 4: Diagnosis results for Scenario 1 using IRT-CNN and Vib-CNN methods

Data source

Average testing accuracy

Standard deviation

CPU time (s)

Vibration signals

71.53% (372/520)

0.5591

470

IRT images

100% (520/520)

0.00

542

The related explanations (highlighted in red) have been added in Section 5.2.2 of page 10 and page 12 in the revised paper.

2. Figure 15 - Please show some examples of the misclassifications and extend discussions on possible noises and errors in the conditions.

Response: Thanks for your comments. As you suggested, the temperature range of 39°C -42°C is selected as an example due to its lowest training and testing accuracies in the revised paper. Fig.16 illustrates the confusion matrix results of proposed IRT-CNN method in the temperature range of 39°C-42°C. As seen in Fig.16, it can be observed that there are 8 testing samples with pitting tooth fault are misclassified into missing tooth fault with the final testing accuracy of 87.69%. This phenomenon can be explained in the following way. Since the temperature distributions of eight health conditions are not stable as the temperature goes up, this will cause the measured IRT images of eight health condition are difficult to be distinguished.

Detailed explanations are added in Section 5.2.3 of page 13. See the sentences highlighted in red. For your convenience, we recall them as follows:

To better understand the classification results under small temperature fluctuations, a temperature range of 39°C -42°C is selected as an example due to its lowest training and testing accuracies. Fig.16 illustrates the confusion matrix results of proposed IRT-CNN method in the temperature range of 39°C-42°C. As seen in Fig.16, it can be observed that there are 8 testing samples with pitting tooth fault are misclassified into missing tooth fault with the final testing accuracy of 87.69%. This phenomenon can be explained in the following way. Since the temperature distributions of eight health conditions are not stable as the temperature goes up, this will cause the measured IRT images of eight health condition are difficult to be distinguished.

Fig. 16 Confusion matrix results with a temperature range of 39°C -42°C.

3. Some English corrections in my reviews. Please consider extensive proof-reading.

(1) 3-a. L26 - Spell out "CM". The abbreviations should be spelled out at the first appearance in body text, not only abstarct.

Response: Thanks. We have added the full name condition monitoring (CM) at the first appearance in the Introduction.  

(2) 3-b. L70 - Spell out "SR", which in not defined before this.

Response: Thanks. We have added the full name softmax regression (SR) at the first appearance in the Introduction.  

(3) 3-c. L124 - "..., which is the followed by a SR ..." - remove "the"

Response: Thanks for your comments. We have conducted the revision exactly as you suggested.
